# Association of Exercise Intensity with the Prevalence of Glaucoma and Intraocular Pressure in Men: A Study Based on the Korea National Health and Nutrition Examination Survey

**DOI:** 10.3390/jcm11164725

**Published:** 2022-08-12

**Authors:** Je-Hyun Seo, Young Lee

**Affiliations:** 1Veterans Medical Research Institute, Veterans Health Service Medical Center, Seoul 05368, Korea; 2Department of Applied Statistics, Chung-Ang University, Seoul 06974, Korea

**Keywords:** ageing, exercise, glaucoma, men, intraocular pressure, physical activity

## Abstract

Adequate exercise is essential for maintaining a healthy lifestyle and preventing ageing-related diseases. The purpose of this study was to assess the associations between exercise and glaucoma, as well as exercise and intraocular pressure (IOP) levels. This study used data from the Korea National Health and Nutrition Examination Surveys 2008–2012, which in total included 10,243 men aged ≥40 years. The presence of glaucoma and the higher IOP of each eye (IOPmax) taken from the health examination survey and the ophthalmic examination were used for analyses. A questionnaire was used to assess exercise activity, which was analysed regarding intensity, frequency, and duration. Regression analyses were used to determine the relationships of exercise parameters with the odds of glaucoma and IOPmax. The prevalence of glaucoma was significantly lower in men who engaged in moderate-to-vigorous intensity exercise compared to those who did not exercise (*p* = 0.012). The odds for glaucoma were the lowest in men engaged in vigorous intensity exercise (*p* = 0.009). However, IOPmax was highest in the vigorous intensity exercise group (*p* = 0.026) with no linear trend pattern. These results suggest that exercise decreased the odds of glaucoma via several factors including non-IOP mechanisms.

## 1. Introduction

Glaucoma, a progressive optic neuropathy, is a leading global cause of irreversible vision loss [1]. It has become a major public health issue in Korea where its prevalence has been steadily increasing [2]. It has been well established that intraocular pressure (IOP) plays a critical role in the development and progression of glaucoma [3]. Other risk factors for glaucoma include age, family history of glaucoma, diabetes mellitus, and genes related to lipid metabolism, such as *ABCA1* and *ELOVL5* [4,5]. In conjunction with IOP-lowering treatments, preventing metabolic deterioration may be a promising therapeutic strategy for glaucoma care. 

Regular physical activity (PA) has been suggested as a safe strategy to counter the general health issues that arise with ageing [6]. To date, the effect of exercise on glaucoma and IOP is poorly understood. Since there are various habitual types of PA, such as yoga, running, and squatting, PA is difficult to quantify, distinct from other risk factors such as diabetes or hypertension. A previous study on the association between exercise and glaucoma using Korean population data demonstrated that vigorous exercise was associated with a higher prevalence of glaucoma (odds ratio [OR] = 1.55, 95% confidence interval [CI]: 1.03−2.33) [7]. However, a subsequent study on the effects of PA on glaucoma using the data of the National Health and Nutrition Examination Survey 2005–2006 in the United States revealed that increased exercise intensity is associated with decreased odds of glaucoma [8]. These controversial results on the association between exercise intensity and glaucoma need to be further elucidated.

Research on the association of PA and IOP may be of interest for glaucoma specialists. Aerobic exercise is known to reduce IOP in the eyes through various mechanisms such as a change in aqueous humour dynamics from altered levels of catecholamine, plasma osmolality, and expansion of Schlemm’s canal dimensions [9,10,11,12]. Some types of PA, such as head-down yoga position, may be related to the increase in IOP [13], whereas other investigators had reported non-significant drops in IOP-related profiles with yoga [9]. Since IOP is a major risk factor for glaucoma, the association between exercise and IOP should also be analysed to understand the association between exercise and glaucoma. Sex-based differences in PA behaviour may affect exercise types and covariates [14], and there is evidence that menopause is a sex-specific risk factor for glaucoma [15]. Moreover, according to one study [7], vigorous PA has a positive association with the prevalence of glaucoma in men (OR = 6.05, 95% CI: 1.67–21.94), whereas another study indicates that vigorous PA protects male runners from glaucoma [16]. Thus, for this research, we opted to study men in the Korea National Health and Nutrition Examination Survey (KNHANES) to reconcile these contradictory findings. Although one prior study evaluated the relationship between exercise and glaucoma using KNHANES data [7], our results may be different due to the difficulty of statistically analysing PA classification and quantification, as well as the extensive analysis and the multiple factors of the KNHANES. Recently, our study group conducted studies to analyse the effects of exercise on osteoporosis [17], metabolic syndrome [18], and sarcopenia [19] based on KNHANES data by reclassifying PA according to exercise intensity, frequency, and duration. These studies demonstrated the feasibility of examining associations with exercise amount in clinical situations using big data analysis. Therefore, the purpose of this study was to investigate the association between exercise and glaucoma, as well as exercise and IOP, for men aged ≥40 years using data from the KNHANES 2008–2012. 

## 2. Materials and Methods

### 2.1. Study Design and Participants

Data from the KNHANES datasets from 2008–2012 produced by the Korea Disease Control and Prevention Agency were used in this study. KNHANES is a nationwide survey with a cross-sectional design used to evaluate the health and nutritional status of the Korean population by assessing medical history, physical examinations, health behaviour surveys, and anthropometric and biochemical measurements. The Institutional Review Board of the Veterans Health Service Medical Center approved the study protocol and waived the requirement of informed consent (IRB No. 2021-10-016) due to the retrospective nature of the study. The study was conducted in compliance with the Declaration of Helsinki. Since gender is highly related to PA, behaviour pattern, and covariates [14], we focused exclusively on men. Hence, we analysed data of 10,243 men aged 40 years or older from KNHANES 2008–2012 (Figure 1). The exclusion criteria were as follows: missing data on PA and eye examination (*n* = 855 and *n* = 209, respectively), conditions that affect PA including chronic disease (stroke, chronic renal failure, and depression), restrictions in the ability to engage in PA (dementia, fracture, etc.), and nutritional issues. We evaluated without filtering outliers since defining the normal range of IOP might lead to selection bias, although the average IOP is 14.91 mmHg [20,21]. Thus, 8343 participants in the KNHANES 2008–2012 were eligible for this study. After excluding missing weight variables (*n* = 1815), 6528 participants were finally analysed in this study (Figure 1).

### 2.2. Assessments of Glaucoma and Intraocular Pressure 

Glaucoma was defined in accordance with the ophthalmologic focus questions (glaucoma-related questions: whether you receive glaucoma treatment at an eye doctor or not) and the examinations in the fourth and fifth KNHANES, which were presented in previous studies [7,22,23] and defined by the International Society of Geographical and Epidemiological Ophthalmology (ISGEO) category I or II diagnostic criteria [24,25], as requiring cup/disc ratio ≥0.6 or cup asymmetry ≥0.2 or disc haemorrhage or against ISNT rule (normal eyes show a characteristic configuration for disc rim thickness of inferior ≥ superior ≥ nasal ≥ temporal) retinal fibre layer defects in fundus photography with/without visual field defect consistent with glaucoma. Using these criteria, subjects were divided into two distinct categories, glaucoma and non-glaucoma. Furthermore, glaucoma stage was classed as preperimetric glaucoma or perimetric glaucoma based on the presence or absence of visual field abnormalities. Furthermore, preperimetric glaucoma was characterized based on the presence or lack of disc alteration and RNFL change. The FDT visual field test results contain a multitude of thresholds, making it difficult to distinguish the stages using the generally used mean deviation. Based on the results of ocular examination, unilateral or bilateral glaucoma were identified. IOP was measured with a Goldmann applanation tonometry (Haag-Streit, AG, Köniz, Switzerland) by eye doctors. In this study, the highest IOP of two eyes was defined as IOPmax, since PA can affect the IOP in both eyes.

### 2.3. Assessment of Physical Activity: Exercise Intensity, Frequency, and Duration

The International Physical Activity Questionnaire Short Form (IPAQ-SF) was used to assess PA. Based on our previously reported classification system [17,18], the questionnaires were classified according to intensity, frequency, and duration. In brief, participants were asked types of PA (no exercise, walking only, moderate intensity exercise, and vigorous intensity exercise) for at least 10 min on a week, duration (how many hours and minutes per day) and frequency (how many days per week). The exercise intensity was categorized as groups with no exercise, walking only, moderate intensity exercise (slow swimming, playing tennis doubles/badminton/volleyball, and moving light objects), and vigorous intensity exercise (running, mountain climbing, soccer/basketball/tennis/squash, fast swimming, fast cycling, rope skipping, and moving heavy objects), which is the classification system described in our previous research [17,18,19]. 

### 2.4. Covariates

The following indexes that may be related to exercise ability and glaucoma were considered among the demographics: age, body mass index (BMI), smoking, alcohol intake, total energy intake, caffeine intake, monthly household income, presence of diabetes, presence of hypertensionpresence of cold hands and feet, refractive errors (spherical equivalents:SE), and IOPmax. For refractive error, an individual may have two SE. For unilateral glaucoma, the SE of the diseased eye was used, while for subject with bilateral glaucoma or without glaucoma, a random SE was used.

### 2.5. Statistical Analyses

The sample weights from the KNHANES data were used in all analyses. Data were expressed as means with standard errors for continuous variables and percentages with standard errors for categorical variables when characterizing participants based on PA intensity. Continuous variables were analysed using the independent t-test or analysis of variance, whereas categorical variables were analysed using the Rao-Scott chi-square test. Subgroup analysis was performed by applying the post-hoc Bonferroni correction after the t-test. Logistic and linear regression analyses were adjusted sequentially for the following variables: unadjusted; adjusted for age and BMI (model 1); model 1 additionally adjusted for smoking, alcohol intake, total energy intake, caffeine intake, monthly household income(model 2); model 2 additionally adjusted for diabetes, hypertension, cold hands and feet (model 3); model 3 additionally adjusted for right spherical equivalents or left spherical equivalents (model 4), and model 4 additionally adjusted for IOPmax (model 5). In each PA intensity group, trend analysis was utilized to examine the presence of glaucoma or IOP in relation to exercise intensity, frequency, and duration. Statistical analyses were performed using the R 3.6.3 program (R Foundation, Vienna, Austria), and the level of statistical significance was set at *p <* 0.05.

## 3. Results

### 3.1. Characteristics of the Study Participants

The mean age of our participants was significantly lower in the group that engaged in vigorous intensity exercise than in the group that performed no exercise (*p* < 0.001, Table 1 and Appendix A). Additionally, the BMI differed by PA intensity group in men (*p* < 0.001). No significant difference in smoking status was observed (*p* = 0.104), but alcohol consumption and monthly income were significantly different between PA and no PA groups (*p* < 0.001 and *p* < 0.001, respectively). There was no difference in refractive errors in both eyes between the PA groups (right eyes *p* = 0.443 and left eyes *p* = 0.949) following adjustment for age and BMI. Total energy intake, total protein intake, and total fat intake were higher in the group engaged in vigorous PA than in the group that did not exercise (*p* < 0.001). The number of cups of caffeine-containing coffee consumed did not differ across the exercise intensity groups (*p* = 0.351). Caffeine consumption has been reported to increase IOP [26]; hence, it was chosen as a variable despite the absence of a statistically significant difference. The prevalence of comorbidities such as hypertension and diabetes were different between the groups (*p* < 0.05), with features such as cold hands and feet showing prevalence differences with non-significant higher significance (*p* = 0.060). Disparities in the duration and frequency of PA were identified between groups according to PA intensity. The laterality of glaucoma characteristics was both eye (*n* = 29, 35.80%), others was unilateral glaucoma (*n* = 52, 64.19%). Although there was no discernible difference in the other variables according to glaucoma stage, the mean age in the perimetric glaucoma group was higher than in that of the preperimetric glaucoma group with RNFL defects and disc abnormality (63.51 ± 2.992 vs. 54.74 ± 2.755, *p* = 0.030, Appendix A).

### 3.2. Associations between Exercise and Glaucoma

The no exercise group exhibited the highest prevalence of glaucoma (2.0%), whereas the vigorous PA group had the lowest prevalence of glaucoma (0.6%) (*p* = 0.012, Table 1). Trend analysis showed that the ORs of glaucoma significantly decreased as the exercise intensity increased in model 2 (*p* = 0.042), model 3 (*p* = 0.034), model 4 (*p* = 0.010), and model 5 (*p* = 0.009) (Figure 2 and Table 2).

For men in the vigorous intensity exercise group, the ORs of glaucoma were 0.233 (95% CI: 0.085–0.637) in model 2, 0.222 (95% CI: 0.080–0.615) in model 3, 0.183 (95% CI: 0.068–0.495) in model 4, and 0.182 (95% CI: 0.067–0.490) in model 5 (Table 2). Men who engaged in moderate intensity exercise also exhibited a lower risk of glaucoma than the no exercise group (OR = 0.338, 95% CI: 0.117–0.974 in model 2; OR = 0.332, 95% CI: 0.114–0.970 in model 3; OR = 0.322, 95% CI: 0.109–0.945 in model 4; and OR = 0.329, 95% CI: 0.112–0.962 in model 5). In the walking only group, the ORs of glaucoma were lower than those in the no exercise group (OR = 0.373, 95% CI: 0.159–0.877 in model 2; OR = 0.351, 95% CI: 0.149–0.826 in model 3; OR = 0.303, 95% CI: 0.126–0.730 in model 4; and OR = 0.306, 95% CI: 0.129–0.727 in model 5). 

When the frequency and duration of exercise were controlled for exercise intensity, there parameters were not related to the prevalence of glaucoma in the exercise intensity groups; only the higher frequency of PA (4–6 days/week) was related to glaucoma (OR = 6.185, 95% CI: 1.516–25.239, *p* < 0.05) in the walking only group (Table 3).

### 3.3. Associations between Exercise and Intraocular Pressure

The IOPmax of the vigorous intensity exercise group was higher than that of the no exercise group (14.86 ± 0.080 vs. 14.38 ± 0.147 mmHg, *p* = 0.003, Table 1). Trend analysis showed that IOPmax significantly differed with increasing exercise intensity in model 1 (*p* = 0.031), model 2 (*p* = 0.025), model 3 (*p* = 0.024), and model 4 (*p* = 0.026) (Figure 2 and Table 2). Men in the vigorous intensity exercise group also showed a higher IOPmax in all models compared to the no exercise group (*p* < 0.05, Table 2). When the frequency and duration of exercise were analysed with the exercise intensity controlled, duration and frequency of exercise were not related to IOPmax in the exercise intensity groups (Table 3 and Figure 3).

## 4. Discussion

Our study showed that in men aged ≥40 years, moderate-to-vigorous intensity exercise was negatively correlated with the odds of glaucoma, whereas IOPmax was not correlated with exercise intensity. It is worth comparing our findings with those of previous studies in relation to the effects of exercise on glaucoma prevention. A previous study using a large prospective cohort of >27,000 males has shown that 10-km runs were related to a reduction in relative glaucoma risk, with a 5% reduction per kilometre per day [16]. An experimental study using aged mice has suggested that exercise by forced swimming significantly improved functional recovery of retinal ganglion cells in an acute IOP elevation model [27]. In addition, recent studies have shown that increased moderate-to-vigorous activity as determined by accelerometer measurements was associated with decreased rates of visual field loss in patients with glaucoma [8,11], which suggests the necessity for further research into the relationship between glaucoma and PA. However, a previous study using South Korean data showed that exercise intensities were associated with increased glaucoma odds [7]. As we used similar datasets (KNHANES 2008–2012) in our study, it is worth evaluating the differences in analysis. One of the main differences is that the classification of PA is different (American College of Sports Medicine [ACSM] vs. categorized PA intensity group). According to ACSM recommendations, healthy adults aged 18–65 years should participate in moderate PA for a minimum of 30 min five days per week, or 20 min of vigorous PA three days per week [28]. The ACSM recommendations represent a standard strategy for assessing the cardiorespiratory endurance of healthy people, which is a high-intensity standard when applied to a Korean cohort, whereas our PA categorization was developed for the classification based on the KNHANES question structure [17,18,19,29]. Previous results are consistent with the more detailed metabolic equivalents (METs) criteria, which showed that regular PA is an important factor in the progression of glaucoma [30].

IOP elevation remains an important risk factor in glaucoma [1]. Several studies have reported that exercise transiently lowers IOP [9,10,12], which is likely to be caused by the type of activity such as strain type, aerobic exercise, intensity, and frequency. However, elevated blood pressure after exercise may contribute to increased IOP due to the production of aqueous humour in the ciliary body and iris [31]. In our study, the IOPmax was highest in the vigorous intensity exercise group without a linear trend pattern. Our findings are supported by previous research demonstrating that fitter persons had higher IOP levels at baseline but a more stable IOP response after resistance training [32,33]. These results suggest that a future selective analysis of the effects on IOP based on different PA types is required. Thus, it is expected that it would be useful to obtain data using continuous IOP monitoring devices [9].

Our results showed that moderate-to-vigorous intensity exercise was negatively correlated with the odds of glaucoma, whereas IOPmax was not. This might suggest that exercise decreased the odds of glaucoma via multiple mechanisms including a non-IOP mechanism. Exercise and PA have whole-body anti-inflammatory effects and modulate neuroinflammation in neurodegenerative diseases, such as Alzheimer’s disease and Parkinson’s disease [34]. Several inflammatory molecules that can influence the optic axon, such as tumour necrosis factor (TNF-α), nitric oxide, and vascular endothelial growth factor [35], are upregulated during glaucoma. TNF-α receptor 1 activity leads to the recruitment of immune cells, causing inflammation and activation of enzymes that induce oxidative stress [36]. It has also been reported that microbiota differences related to inflammation are observed in glaucoma [37,38]. In addition, a recent study showed that TNF-α triggers sterile alpha and Toll/interleukin-1 receptor motif-containing 1 (*SARM1*)-dependent axon degeneration, oligodendrocyte loss, and subsequent retinal ganglion cell death [39]. A potential link may exist for glaucoma pathogenesis via axonal survival factors, such as nicotinamide mononucleotide adenylyltransferase 2 (*NMNAT2*) and stathmin 2 (*STMN2*) [39]. Regarding neuroprotection, studies have shown that exercise protected against glaucoma through brain-derived neurotrophic factor (BDNF) signalling [40,41], which showed that BDNF levels were significantly reduced in the injured retinas of non-exercised mice but maintained in exercised mice. This may explain the mechanism by which exercise protects against glaucoma. 

A major advantage of this study is that it includes a large representative population with weighted data that reflects nationwide prevalence, estimates PA amount based on exercise intensity, duration, and frequency [17,18], and uses ophthalmologic focus questions (glaucoma-related questions), as well as ocular examination and IOP data measured by an eye doctor. In addition, we did not construct an exercise program to analyse its effectiveness; rather, we divided exercise patterns based on the validity of the IPAQ-SF in Koreans. Nevertheless, this study has several limitations. First, this was a cross-sectional study. Thus, our results were not able to identify causal relationships. Our results also were not able to account for those unable to exercise vigorously due to having glaucoma or visual field defects. Patients with glaucoma and systemic disease may not find it easy to exercise. Glaucoma patients routinely performed less exercise than age-matched controls [42]. Conversely, there is a possibility that glaucoma patients might use more aerobic exercise to overcome glaucoma. In our study, it is crucial to consider the walking-only group’s high frequency of PA (4–6 days per week) was associated with glaucoma (OR = 6.185) and what this implies. It makes more sense to assume that glaucoma patients regularly engage in low-intensity exercise owing to visual impairment than that low-intensity exercise causes glaucoma. However, in our study, the effects of these factors may be reduced since this adjustment is included in the statistics.

Second, since all the information was based on self-reported health surveys, there is the potential for recall or acquiescence bias, which could lead to misclassification. Although evaluating normality is critical for data analysis, because KNHANES dataset is a ‘complex sampling design’, we choose to show the results using histograms (Appendix A) and non-parametric tests (Appendix A). In addition, central corneal thickness is an important mediating factor for IOP measurement; however, since the corneal pachymetry test was not included in the KNHANES dataset, the failure to correct IOP findings should be taken into account when interpreting the results. In our study, the vigorous exercise group had lower IOP, although all IOP levels were within normal ranges. Diunal IOP fluctuation is one of the risk factors for glaucoma [43], but in this investigation, IOP was measured only once, and time-dependent changes were not analysed. Since the influence of diurnal IOP fluctuation was not taken into consideration, any interpretation should be conducted with caution. Third, the relationship between specific exercises and glaucoma may have been estimated incorrectly in our analyses, which were categorized based on general exercise intensity. Excluding exercise that is known to negatively influence glaucoma, as it would be an arbitrary criterion, might also affect our results. Thus, these findings warrant confirmation by further prospective studies. Fourth, variables were not examined using multivariate statistics, and the frequency and duration of each exercise were assessed using categorization rather than linear analysis, which may be arbitrary. With around 6500 participants, it is possible to analyse the KNHANES dataset using a multivariate model. However, the number of glaucoma patients was restricted to 81, constraining the study; hence, a feasible model was developed by prioritizing glaucoma risk factor variables. Although the categorizations of exercise frequency and duration were rather arbitrary, it was a strategy to create a suitable model since the number of glaucoma patients was small. These aspects should be taken into account when interpreting the results. Fifth, this study did not pay attention to gender, despite a prior study that examined gender differences in the correlation analysis between PA and glaucoma [7]. However, since gender is a significant factor in science and some researchers may be interested in learning how exercise affects men and women differently on the glaucoma and IOP. For this issue, the results were attached the analysis for women (Appendix A).

## 5. Conclusions

Exercise intensity was negatively correlated with odds of glaucoma with trend patterns. Logistic regression analysis revealed a strong relationship between exercise intensity and decreased odds of glaucoma, suggesting that exercise may have protective effects against glaucoma. While IOPmax was higher in the vigorous intensity exercise group compared to the no exercise group, a trend analysis of IOPmax with exercise intensity did not show IOPmax increasing with exercise intensity. These results might suggest that exercise decreased the odds of glaucoma through various mechanisms including a non-IOP mechanism, as well as IOP fluctuation effects. Further studies are warranted to investigate the association between exercise and glaucoma.

## Figures and Tables

**Figure 1 jcm-11-04725-f001:**
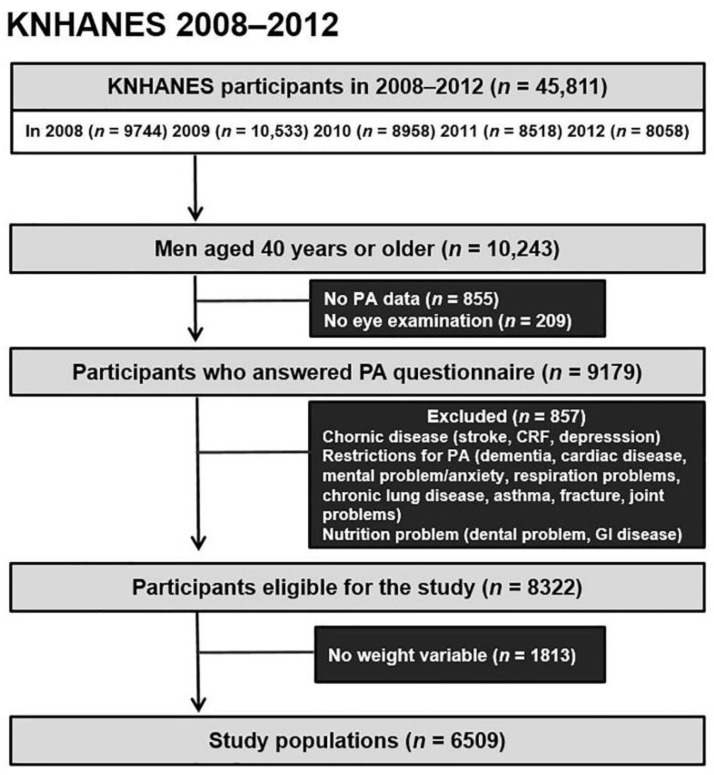
Flowchart of the study population. CRF, chronic renal failure; GI, gastrointestinal; KNHANES, Korea National Health and Nutrition Examination Survey; PA, physical activity.

**Figure 2 jcm-11-04725-f002:**
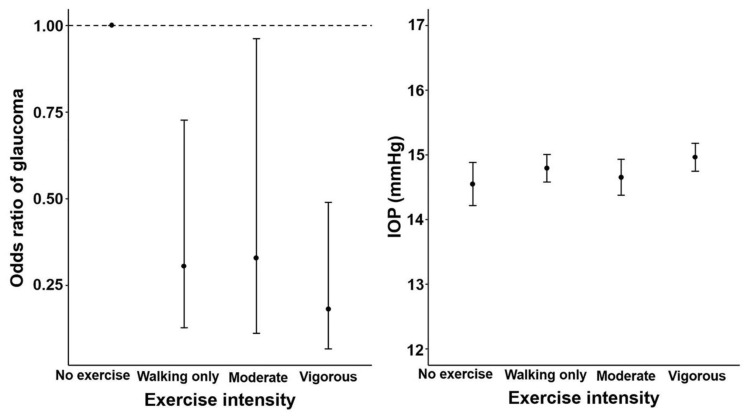
Odds ratios for glaucoma and mean IOP according to the intensity of exercise in men aged ≥40 years. Trend *p* using a logistic and linear regression model after adjusting for age, body mass index, smoking, drinking, monthly income, total energy intake, caffeine intake, hypertension diabetes mellitus, spherical equivalents. For odds of glaucoma, model 5 was used and for IOP, model 4 was used. Error bars indicate 95% confidence intervals. IOP, intraocular pressure.

**Figure 3 jcm-11-04725-f003:**
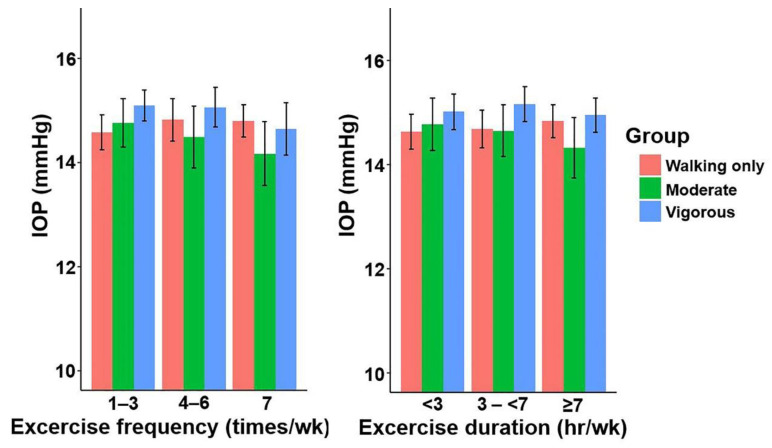
Mean IOPmax according to the frequency and duration of physical activity in men aged ≥40 years. Linear regression model after adjusting for age, body mass index, smoking, drinking, monthly income, total energy intake, caffeine intake and diabetes mellitus, hypertension and spherical equivalents. Error bars indicate 95% confidence intervals. IOPmax, the highest intraocular pressure of both eyes.

**Table 1 jcm-11-04725-t001:** Baseline characteristics of study subjects.

	Total	No Exercise	Walking Only	Moderate Intensity Exercise	Vigorous Intensity Exercise	
	(*n* = 6528)	(*n* = 651)	(*n* = 2337)	(*n* = 1132)	(*n* = 2408)	*p*
Age, years	54.16 ± 0.169	55.25 ± 0.568	57.31 ± 0.297	54.38 ± 0.385	51.39 ± 0.200	<0.001
		a	b	a	c	
BMI, kg/m^2^	24.08 ± 0.048	23.76 ± 0.155	23.90 ± 0.086	23.88 ± 0.11	24.37 ± 0.075	<0.001
		a	a	a	b	
SE Right, D	−0.59 ± 0.033	−0.58 ± 0.109	−0.43 ± 0.059	−0.52 ± 0.072	−0.74 ± 0.048	<0.001
		ab	a	ab	b	
age adjusted		−0.33 ± 0.103	−0.32 ± 0.053	−0.22 ± 0.065	−0.22 ± 0.043	0.464
age and BMI adjusted		−0.33 ± 0.103	−0.32 ± 0.053	−0.22 ± 0.065	−0.22 ± 0.044	0.443
SE left, D	−0.54 ± 0.032	−0.46 ± 0.101	−0.33 ± 0.056	−0.51 ± 0.075	−0.73 ± 0.048	<0.001
		ab	a	ab	b	
age adjusted		−0.18 ± 0.092	−0.23 ± 0.05	−0.2 ± 0.066	−0.19 ± 0.044	0.954
age and BMI adjusted		−0.18 ± 0.092	−0.23 ± 0.05	−0.2 ± 0.067	−0.19 ± 0.044	0.949
IOPmax, mmHg	14.68 ± 0.061	14.38 ± 0.147	14.62 ± 0.090	14.51 ± 0.118	14.86 ± 0.080	0.003
		a	ab	ab	b	
age adjusted		14.34 ± 0.146	14.60 ± 0.089	14.46 ± 0.120	14.76 ± 0.084	0.016
age and BMI adjusted		14.36 ± 0.146	14.60 ± 0.089	14.47 ± 0.119	14.75 ± 0.084	0.031
Glaucoma, %	0.9 (0.12)	2.0 (0.71)	1.1 (0.22)	0.7 (0.27)	0.6 (0.15)	0.012
Alcohol consumption, %						<0.001
None	16.9 (0.53)	19.2 (1.75)	21.5 (1.07)	16.7 (1.35)	12.9 (0.79)	
Moderate	40.1 (0.79)	41.7 (2.52)	36.8 (1.26)	36.4 (1.86)	43.7 (1.22)	
Heavy	42.9 (0.78)	39.0 (2.28)	41.6 (1.34)	46.8 (1.92)	43.3 (1.19)	
Smoking status, %						0.104
Never	15.1 (0.55)	14.1 (1.69)	15.4 (0.95)	15.1 (1.24)	15.1 (0.85)	
Ex-	45.8 (0.81)	41.4 (2.43)	46.6 (1.31)	43.1 (1.88)	47.3 (1.27)	
Current	39.1 (0.80)	44.5 (2.48)	38.0 (1.30)	41.8 (1.86)	37.6 (1.22)	
Monthly household income, %						<0.001
Lowest	15.8 (0.58)	19.9 (1.66)	21.5 (1.05)	15.9 (1.26)	10.3 (0.74)	
Medium-lowest	25.2 (0.79)	30.5 (2.37)	26.9 (1.25)	23.2 (1.59)	23.5 (1.13)	
Medium-highest	28.4 (0.73)	29.7 (2.30)	27.4 (1.24)	26.8 (1.66)	29.5 (1.14)	
Highest	30.6 (0.86)	19.8 (2.01)	24.2 (1.17)	34.0 (1.84)	36.8 (1.34)	
Intake caffeine, cup/day %						0.351
<1	24.7 (0.70)	24.1 (2.09)	25.5 (1.27)	25.7 (1.70)	24.0 (1.07)	
1	19.7 (0.64)	18.7 (2.03)	20.9 (1.11)	18.6 (1.45)	19.5 (0.99)	
2	23.8 (0.71)	19.9 (2.11)	23.5 (1.24)	23.8 (1.66)	24.8 (1.11)	
≥3	31.8 (0.84)	37.3 (2.54)	30.1 (1.46)	31.9 (1.87)	31.7 (1.24)	
Total energy intake, kcal/d	2326.96 ± 15.316	2250.29 ± 42.606	2196.76 ± 23.698	2372.25 ± 37.957	2427.73 ± 23.133	<0.001
		ab	a	bc	c	
Hypertension, %	24.5 (0.65)	22.7 (1.94)	29.2 (1.18)	25.0 (1.57)	21.2 (0.94)	<0.001
Diabetes, %	10.3 (0.43)	9.4 (1.28)	13.4 (0.84)	11.0 (1.18)	7.9 (0.59)	<0.001
Cold hands and feet, %	11.8 (0.53)	14.1 (1.71)	12.8 (0.87)	9.6 (1.04)	11.3 (0.79)	0.060

Data with the same lowercase letters indicate non-specific differences between groups, while those with different letters are statistically different, based on post hoc test Data are expressed as the mean ± standard errors or the percentage.

**Table 2 jcm-11-04725-t002:** Regression analysis for glaucoma and IOPmax according to intensity of physical activity.

	No Exercise	Walking Only	Moderate Intensity Exercise	Vigorous Intensity Exercise	*p* ^‡^
	*n* = 651	*n* = 2337	*n* = 1132	*n* = 2408	
**Glaucoma**					
Unadjusted	1	0.559 (0.253–1.233)	0.360 (0.131–0.990) *	0.291 (0.123–0.687) †	0.027
Model 1	1	0.527 (0.236–1.176)	0.396 (0.139–1.130)	0.380 (0.144–1.005)	0.235
Model 2	1	0.373 (0.159–0.877) *	0.338 (0.117–0.974) *	0.233 (0.085–0.637) †	0.042
Model 3	1	0.351 (0.149–0.826) *	0.332 (0.114–0.970) *	0.222 (0.080–0.615) †	0.034
Model 4	1	0.303 (0.126–0.730) †	0.322 (0.109–0.945) *	0.183 (0.068–0.495) †	0.010
Model 5	1	0.306 (0.129–0.727) †	0.329 (0.112–0.962) *	0.182 (0.067–0.490) †	0.009
**IOPmax**					
Unadjusted	14.38 ± 0.147	14.62 ± 0.09	14.51 ± 0.118	14.86 ± 0.08 †	0.003
Model 1	14.36 ± 0.146	14.60 ± 0.089	14.47 ± 0.119	14.75 ± 0.084 *	0.031
Model 2	14.34 ± 0.154	14.62 ± 0.095	14.43 ± 0.124	14.75 ± 0.088 *	0.025
Model 3	14.53 ± 0.170	14.79 ± 0.110	14.62 ± 0.139	14.94 ± 0.109 *	0.024
Model 4	14.55 ± 0.170	14.79 ± 0.110	14.65 ± 0.141	14.96 ± 0.109 *	0.026

Unadjusted: no adjustment; model 1: adjusted by age and BMI; model 2: model 1+ smoking, alcohol intake, total energy intake, caffeine intake, monthly household income; model 3: model 2+ diabetes, hypertensioncold hand and foot; model 4: model 3+ right spherical equivalents or left equivalents; model 5: model 4+ IOPmax Glaucoma expressed in 95% confidence intervals and IOP was summarized mean ± standard errors, *: indicate, if *p* < 0.05, †: indicate, if *p* < 0.01 compared with no exercise group, *p*
^‡^ value for trend *p* value.

**Table 3 jcm-11-04725-t003:** Regression analysis of glaucoma and intraocular pressure according to the frequency or duration of physical activity in men.

	Men (*n* = 6528)	
	Glaucoma ^§^	IOP Max ^#^
Walking only	*n* = 2337	
Frequency		
1–3	1	14.58 ± 0.171
4–6	6.185 (1.516–25.239) *	14.82 ± 0.207
everyday	2.468 (0.707–8.611)	14.8 ± 0.160
*p*	0.030	0.419
Duration		
<3	1	14.63 ± 0.172
3- <7	1.080 (0.323–3.614)	14.69 ± 0.184
≥7	2.028 (0.606–6.794)	14.84 ± 0.161
*p*	0.374	0.556
Moderate Intensity Exercise	*n* = 1132	
Frequency		
1–3	1	14.76 ± 0.237
4–6	0.648 (0.117–3.589)	14.49 ± 0.300
everyday	0.395 (0.045–3.432)	14.17 ± 0.312 *
*p*	0.673	0.090
Duration		
<3	1	14.77 ± 0.257
3–<7	1.156 (0.265–5.045)	14.65 ± 0.253
≥7	0.210 (0.031–1.439)	14.32 ± 0.297
*p*	0.237	0.263
Vigorous intensity Exercise	*n* = 2408	
Frequency		
1–3	1	15.10 ± 0.152
4–6	0.711 (0.166–3.049)	15.06 ± 0.195
everyday	1.015 (0.168–6.150)	14.64 ± 0.257
*p*	0.899	0.215
Duration		
<3	1	15.02 ± 0.173
3–<7	0.541 (0.152–1.917)	15.16 ± 0.174
≥7	0.278 (0.034–2.308)	14.95 ± 0.167
*p*	0.360	0.499

*: indicate, if *p* < 0.05 compared with 1–3 in exercise frequency, <3 in exercise duration, ^§^ Logistic regression analysis for glaucoma was adjusted with age, BMI, smoking, alcohol intake, caffeine intake, total energy intake, monthly household income, diabetes, hypertension, cold hand and foot, right spherical equivalents or left equivalents, and IOPmax, ^#^ Linear regression analysis for IOP was age, BMI, smoking, alcohol intake, caffeine intake, total energy intake, monthly household income, diabetes, hypertension, cold hand and foot, right spherical equivalents or left equivalents.

## Data Availability

The datasets used and/or analysed in this study are available on the official website of the Korea National Health & Nutrition Examination Survey (KNHANES) (https://knhanes.kdca.go.kr/knhanes/eng/index.do (accessed on October 2021)).

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
