# Peer review of "Association of Exercise Intensity with the Prevalence of Glaucoma and Intraocular Pressure in Men: A Study Based on the Korea National Health and Nutrition Examination Survey"

_jcm, 2022, doi:10.3390/jcm11164725_

Round 1

Reviewer 1 Report

Association of exercise intensity with the prevalence of glaucoma and intraocular pressure in men: A study based on the Korea National Health and Nutrition Examination Survey

The manuscript is well-written. The main criticism would be that classification of physical activity was unclear and might not be appropriate.

1.     The authors classified the intensity group based on previous week's activity, which could not represent 'regular physical activity’. However, when the frequency of PA in a week was taken into account (which better reflects regularity), walking 4-6 times per week had a significantly higher OR (OR 6.1) than lower frequency. The matter should be brought up in the discussion.

2.     Patients with outliner IOP ranges were excluded by the authors. One of the reasons was that they could be related to untrustworthy values and measurement errors. Another exclusion criterion was to exclude IOPs greater than 25 mmHg. It was unclear why the authors had to exclude those with IOPs greater than 25 mmHg. Furthermore, the study included two outcomes: glaucoma and IOP max. The main outcome was glaucoma, as defined by a questionnaire. Outliers in IOP, whether due to measurement or other factors, were not associated with self-reported glaucoma. These exclusion criteria may not be required for the glaucoma outcome analysis. Excluding these patients resulted in unneeded selection bias.

3.     Section 2.2 (Line 102-111): Because glaucoma is the focus of the study, it is suggested that more information about how glaucoma was determined be provided.

4.     Section 2.3 (Line 113-126): This section is critical because it deals with categorizing variables of interest. It was still unclear how the authors classified exercise intensity. "Participants were asked if they had participated in at least 10 minutes of various types of PA for exercise in the previous week." Was the category based solely on physical activity from the previous week? It is strongly advised that the authors clarify this section.

5.     Why did the author choose arthritis as one of the covariates?

6.     Minor correction: The first row of the table on Page 5 should be moved to Page 4 because it demonstrated the post-hoc analysis of the previous row, which was on Page 4.

Author Response

Comments and Suggestions for Authors

Association of exercise intensity with the prevalence of glaucoma and intraocular pressure in men: A study based on the Korea National Health and Nutrition Examination Survey

The manuscript is well-written. The main criticism would be that classification of physical activity was unclear and might not be appropriate.

 → We appreciate the feedback. We have rewritten the manuscript according to the reviewer suggestions. Our answers to each comment are provided in a point-by-point manner in blue lighted or change mark.

  1. The authors classified the intensity group based on previous week's activity, which could not represent 'regular physical activity’. However, when the frequency of PA in a week was taken into account (which better reflects regularity), walking 4-6 times per week had a significantly higher OR (OR 6.1) than lower frequency. The matter should be brought up in the discussion.

→ We appreciate the nice comments. It was corrected due to an error in the questionnaire's translation which was a questionnaire that includes the meaning of ‘regular PA’ in Korean. (line 126-129).

asked types of PA (no exercise, walking only, moderate intensity exercise, and vigorous intensity exercise) for at least 10 minutes on a week, duration (how many hours and minutes per day) and frequency (how many days per week). The exercise intensity was categorized as”

The answer for second comment was added to the section for discussion. (line 320-324). “overcome glaucoma. In our study, it is crucial to consider what it implies that the walking-only group's high frequency of PA (4-6 days per week) was associated with glaucoma (OR=6.185). It makes more sense to assume that glaucoma patients regularly engage in low-intensity exercise owing to visual impairment than that low-intensity exercise causes glaucoma. However”

  1. Patients with outliner IOP ranges were excluded by the authors. One of the reasons was that they could be related to untrustworthy values and measurement errors. Another exclusion criterion was to exclude IOPs greater than 25 mmHg. It was unclear why the authors had to exclude those with IOPs greater than 25 mmHg. Furthermore, the study included two outcomes: glaucoma and IOP max. The main outcome was glaucoma, as defined by a questionnaire. Outliers in IOP, whether due to measurement or other factors, were not associated with self-reported glaucoma. These exclusion criteria may not be required for the glaucoma outcome analysis. Excluding these patients resulted in unneeded selection bias.

→ We appreciate the feedback. At first, the authors considered it more reasonable to remove the outlier IOP, however, these concerns were raised with the two reviewers (selection bias). The IOP measured by eye doctor by Goldmann on examination day. Glaucoma defined by both questionnaire and fundus photography. In reviewers’ opinion, the authors agree that controlling of IOP is not really necessary and might lead misunderstanding. Hence, the analysis was performed without removing of the outlier IOP and re-analyze the dataset accordingly (Fig 1, Fig 2, Fig 3 Table 1, Table 2, Table 3).

  1. Section 2.2 (Line 102-111): Because glaucoma is the focus of the study, it is suggested that more information about how glaucoma was determined be provided.

→ We appreciate the feedback. We have made the correction in details glaucoma definitions and supporting eye examination (line 103-118). Another reviewer advised that each glaucoma stage be studied and presented for bilateral glaucoma, which led to different results (Table S2, lines184-188).

  1. Section 2.3 (Line 113-126): This section is critical because it deals with categorizing variables of interest. It was still unclear how the authors classified exercise intensity. "Participants were asked if they had participated in at least 10 minutes of various types of PA for exercise in the previous week." Was the category based solely on physical activity from the previous week? It is strongly advised that the authors clarify this section.

→ We appreciate the feedback. We have made the corrections as suggested (line 126-130).

  1. Why did the author choose arthritis as one of the covariates?

→ We appreciate the feedback. Initially, a model was developed that incorporated factors in descriptive statistics as well as glaucoma risk factors and physical activity factors. Arthritis itself is related to aerobic exercise and exercise intensity and some difference according exercise –intensity groups (P<0.05). But, this analysis may have confused readers and no relation with eye condition, thus unnecessary covariates (such as arthritis) were deleted and the data was evaluated again (line 140-145). We have made the correction as suggested. In addition, refractive errors were something confusing for 2 variables from single person. (line 145-147)

  1. Minor correction: The first row of the table on Page 5 should be moved to Page 4 because it demonstrated the post-hoc analysis of the previous row, which was on Page 4.

→ We appreciate the feedback. We have made the correction tables into Table 1 and Table S1 for better understanding (Table 1, Table S1).

Reviewer 2 Report

First, thanks for this nice work!

I, though, have few comments:

L11: add "levels" after IOP 

L21: it is true that you demonstrated that lower IOP in vigorous exercise group but all IOP levels were within normal limit. And your study did not consider factors like diurnal IOP fluctuation levels. Then I would be cautious here. 

Further, it would be helpful to see the effects of exercise on the glaucoma patients in different exercise groups.

L56-58: Now your study confirmed one observation that vigorous PA protects male runners from glaucoma and contradicts another. one difference you mentioned is that you did not include patients with IOP higher than 25 mmHg. Do you think including this group will affect the results? It would be more convincing to see the results especially as I am not fully convinced removing of IOP outliers.

L102-110: Glaucoma characterization needs further details: Which glaucoma type, stage of damage especially in each exercise group. Do all glaucoma subjects have bilateral damage?

At which time was the eye check done in relation to questionnaire?

 L133: Did authors run normality check on data? Specify please!

L141-145 and L127-132: In the adjusted models you specified, please include those relevant to glaucoma or cite studies to confirm the covariates you specified. I would stick to the most well-known and influential covariates. E.g., I don't understand model 4 since you only chose only one eye but you adjusted for both eyes’ refraction. 

L169: Table 1 is long and it would be more informative if authors focus variables related to glaucoma. Authors might put the rest of variables in a supplementary file. 

L216: Replace "marginal significance"> a non-significant trend of higher IOP. 

L238-240: In this study exercise and rates of visual field loss were not investigated and the sentence L238-240 here then does not apply.

Finally, gender in science is very important and I would not exclude women from this study and I would rather compare and investigate the effects of exercise between males and females.  

Author Response

Comments and Suggestions for Authors

First, thanks for this nice work!

I, though, have few comments:

→ We appreciate the feedback. We have rewritten the manuscript according to all the reviewer suggestions. Our answers to each comment are provided in a point-by-point manner in blue lighted or change mark.

L11: add "levels" after IOP 

→ We appreciate the feedback. We added “levels” after IOP (line 11).

L21: it is true that you demonstrated that lower IOP in vigorous exercise group but all IOP levels were within normal limit. And your study did not consider factors like diurnal IOP fluctuation levels. Then I would be cautious here. Further, it would be helpful to see the effects of exercise on the glaucoma patients in different exercise groups.

→ We appreciate the feedback. According to the reviewer's assessment, the interpretation of data requires cautions. This is included in the limitations (line 343-346).

In our study, the vigorous exercise group had lower IOP, although all IOP levels were within normal ranges. Diunal IOP fluctuation is one of the risk factor for glaucoma [43] but in this investigation, IOP was measured only once, and time-dependent changes were not analysed. Since the influence of diurnal IOP fluctuation was not taken into consideration, interpretation should be done with caution.

L56-58: Now your study confirmed one observation that vigorous PA protects male runners from glaucoma and contradicts another. one difference you mentioned is that you did not include patients with IOP higher than 25 mmHg. Do you think including this group will affect the results? It would be more convincing to see the results especially as I am not fully convinced removing of IOP outliers.

→ We appreciate the feedback. the authors considered it more reasonable to remove the outlier IOP, however, these concerns were raised with the two reviewers (selection bias). Hence, the new analysis was performed without removing of the outlier IOP, which might be arbitrary. We revised statistics analysis again and entire results (Table 1, 2, 3, Figure 1,2,3)

L102-110: Glaucoma characterization needs further details: Which glaucoma type, stage of damage especially in each exercise group. Do all glaucoma subjects have bilateral damage?

→ We appreciate the feedback. We added definition of glaucoma in detail in this study (lines 103-118). Bilateral damage (n=29, 35.80%) Table S2 was suggested for descriptive parameters according to glaucoma damage. The authors would appreciate it if reviewer could understand that, unlike glaucoma evaluation (OCT, SAP) in a clinic, there is a limit to analysis using only the information adopted from survey data.

At which time was the eye check done in relation to questionnaire?

 L133: Did authors run normality check on data? Specify please!

→ We appreciate the feedback.

1) Normality was not checked because the data we used was a large sample with a “Complex sampling design”. For this issue, we provided non-parametric results in the Table S3.

2) Since IOP is used as a result variable and displays a histogram, it may be utilized with linear regression without any issues because it is typically symmetrical and resembles a normal distribution as Figure S1 (Histrogram).

(lines 337-340)

“Although evaluating normality is critical for data analysis, because KNHANES dataset is a ‘complex sampling design’, we choose to show the results using histograms (Figure S1) and non-parametric tests (Table S3).”

L141-145 and L127-132: In the adjusted models you specified, please include those relevant to glaucoma or cite studies to confirm the covariates you specified. I would stick to the most well-known and influential covariates. E.g., I don't understand model 4 since you only chose only one eye but you adjusted for both eyes’ refraction. 

→ We appreciate the feedback. Initially, a model was developed that incorporated factors in descriptive statistics as well as glaucoma risk factors and activity factors, but this analysis may have confused readers, thus unnecessary covariates were deleted and the data was evaluated again. We have made the correction as suggested (lines 141-147, lines 156-161).

L169: Table 1 is long and it would be more informative if authors focus variables related to glaucoma. Authors might put the rest of variables in a supplementary file. 

→ We appreciate the feedback. We have made the correction for focus variables (adjustment factors) and entire table (Table S1).

L216: Replace "marginal significance"> a non-significant trend of higher IOP. 

→ We appreciate the feedback. According to the results of recent analysis, all variables were significant that expression was changed since all meets P<0.05 (lines 236-239).

Instead, we replaced the “marginal significance” to a non-significant expressions (former: marginal significance) (line 182).

L238-240: In this study exercise and rates of visual field loss were not investigated and the sentence here then does not apply.

→ We appreciate the feedback. The authors agree with the comments. We have made the correction as suggested (Lines 262-263)

which suggests the necessity for further research into the relationship between glaucoma and PA.”

Finally, gender in science is very important and I would not exclude women from this study and I would rather compare and investigate the effects of exercise between males and females.  

→ We appreciate the feedback. The comparison of gender difference in glaucoma and PA has already been studied (PLoS One 12: e0171441 DOI 10.1371/journal.pone.0171441), and our study focused on introducing the meaning of IOP here. Because the study in that instance was so difficult, I attempted to concentrate on the analysis of Men data. However, we believe that some readers may be interested in the outcomes of women as well. Hence, for better understanding, we attached the analysis of PA and glaucoma, IOP in women for Table S4.

Round 2

Reviewer 2 Report

Thanks a lot!

Authors have fully addressed my comments/suggestions.